# Novel Benzothiazole Boc-Phe-Phe-Bz Derivative Dipeptide Forming Fluorescent and Nonlinear Optical Self-Assembled Structures

**DOI:** 10.3390/molecules30040942

**Published:** 2025-02-18

**Authors:** Rosa M. F. Baptista, Daniela Santos, N. F. Cunha, Maria Cidália R. Castro, Pedro V. Rodrigues, Ana V. Machado, Michael S. Belsley, Etelvina de Matos Gomes

**Affiliations:** 1Centre of Physics of Minho and Porto Universities (CF-UM-UP), Laboratory for Physics of Materials and Emergent Technologies (LaPMET), University of Minho, Campus de Gualtar, 4710-057 Braga, Portugal; id11280@alunos.uminho.pt (D.S.); ncunha@fisica.uminho.pt (N.F.C.); belsley@fisica.uminho.pt (M.S.B.); 2Institute for Polymers and Composites, University of Minho, Campus de Azurém, 4800-058 Guimarães, Portugal; cidaliacastro@dep.uminho.pt (M.C.R.C.); pedro.rodrigues@dep.uminho.pt (P.V.R.); avm@dep.uminho.pt (A.V.M.)

**Keywords:** diphenylalanine dipeptides, benzothiazole, self-assembling, optical properties

## Abstract

This work explores the self-assembly and optical properties of a novel chiral, aromatic-rich Boc-Phe-Phe dipeptide derivative functionalized with a benzothiazole bicyclic ring that forms supramolecular structures. Leveraging the well-known self-assembling capabilities of diphenylalanine dipeptides, this modified derivative introduces a heterocyclic benzothiazole unit that significantly enhances the fluorescence of the resulting nanostructures. The derivative’s rich aromatic character drives the formation of supramolecular structures through self-organization mechanisms influenced by quantum confinement. By adjusting the solvent system, the nanostructures exhibit tunable morphologies, ranging from nanospheres to nanobelts. The nonlinear optical properties of these self-assembled structures were studied and an estimated $d_{eff}$ of ~0.9 pm/V was obtained, which is comparable to that reported for the highly aromatic triphenylalanine peptide.

## 1. Introduction

Self-assembly into a variety of superstructures, both in solution and in the solid state, is a key feature of peptides, resulting from the self-organization of their constituent amino acid building blocks [1,2]. One of the most frequently used building blocks in peptide synthesis is chiral phenylalanine (Phe) in either the L or D form, a precursor amino acid in the formation of several essential amino acids. The molecular aromatic dipeptides diphenylalanine (L-phenylalanine-L-phenylalanine-OH, hereafter referred to as Phe-Phe) and its derivative, *N-tert*-butoxycarbonyl-protected diphenylalanine (Boc-L-phenylalanine-L-phenylalanine-OH, hereafter referred to as Boc-Phe-Phe) are short dipeptides comprising two naturally occurring L-phenylalanine residues. Their exceptional mechanical, linear, and nonlinear optical properties, as well as their energy harvesting properties enabled by the piezoelectric effect, have attracted considerable and extensive research interest [3,4,5,6,7,8].

Phe-Phe is a widely studied dipeptide that self-assembles into a variety of well-ordered structures such as nanotubes [9], nanowires [10], and nanorods [11]. Similarly, Boc-Phe-Phe forms highly ordered tubular nanostructures and nanospheres by self-assembling under different dissolution conditions [9,10,11,12,13]. These dipeptides also play a significant role in the formation of nanoparticles under specific conditions, as supported by the recent literature [14].

Their rich hydrogen bonding network together with directional intermolecular π–π interactions enables their self-assembling with quantum nanocrystalline regions under the form of quantum dots with strong quantum confinement [11]. This phenomenon manifests itself as strong fluorescence, making dipeptide superstructures promising materials for photonic applications [11]. Because all amino acids occur naturally as chiral molecular identities, apart from glycine, they further crystallize with acentric symmetry. The lack of a center of symmetry means that the corresponding crystals display properties such as piezoelectricity and optical second harmonic generation. Therefore, dipeptides formed from chiral amino acids are attractive bio-systems for exploring both molecular and crystal properties [15,16,17,18].

It has been demonstrated that dipeptides with an increased aromatic group content relative to Boc-Phe-Phe may result in materials with enhanced optical and mechanical properties, particularly fluorescence and rigidity [17].

Based on these results, we decided to increase the aromaticity of Boc-Phe-Phe by attaching a benzothiazole (Bz) group to its carboxylic acid (COOH) group. Benzothiazoles are heterocyclic compounds with a bicyclic ring, forming a planar rigid structure and π-electron conjugated system, which is a preferred structure that exhibits strong fluorescence both in the solid state and in solutions, and its derivatives have recently been used to construct fluorescent probes and sensors [19,20]. In addition, benzothiazole derivatives have found a wide range of applications, from pharmaceuticals to nonlinear optics [21,22].

Here, we describe the synthesis and chemical and thermal characterization of a new dipeptide, *N-tert*-butoxycarbonyl-protected diphenylalanine Boc-L-phenylalanine-L-phenylalanine-benzothiazole (hereafter Boc-Phe-Phe-Bz). Its self-assembly in different solvents and linear and nonlinear optical properties are also described.

## 2. Results and Discussion

### 2.1. Synthesis

The synthetic procedure used to prepare this compound is illustrated in Figure 1. A simple coupling reaction between 2-amino-6-ethoxybenzothiazole (Bz-NH_2_) and Boc-diphenylalanine (Boc-Phe-Phe) produced Boc-Phe-Phe-Bz in a good yield of 68%. The synthesis of Boc-Phe-Phe was first carried out by protecting the carboxylic acid terminal of phenylalanine by reacting it with thionyl chloride in methanol, obtaining the corresponding methyl ester of the amino acid. Subsequently, the amino terminal was protected by a reaction with di-*tert*-butylpyrocarbonate (Boc_2_O), giving rise to the N-Boc-protected amino acid. Boc-Phe-Phe was obtained by liquid-phase synthesis through the coupling of the phenylalanine methyl ester with the N-Boc protected amino acid, using DCC (dicyclohexylcarbodiimide)/HOBt (1-hydroxybenzotriazole) as coupling agents. Deprotection of the carboxylic acid terminal was achieved by saponification with NaOH 2 M in methanol at room temperature. More details on the synthetic procedure are given in a previously published article [3].

The reaction of Bz-NH_2_ with Boc-Phe-Phe to obtain Boc-Phe-Phe-Bz under normal amide coupling conditions, using DCC/HOBt as coupling agents, produced the compound in a 68% yield. The final dipeptide compound was fully characterized (see Appendix A) through ^1^H and ^13^C NMR spectroscopy (Appendix A), Fourier transform infrared spectroscopy analysis (Appendix A), differential scanning calorimetry (DSC), and thermal gravimetric analysis (TGA) (Appendix A). Thermal analysis indicated that the dipeptide initiated the degradation process at 222 °C.

### 2.2. Linear Optical Properties: Absorption and Fluorescence

Quantum electron-optical confinement (QC) was reported for Boc-Phe-Phe self-assembled nanotubes (NTs) in ethanol, manifesting in the absorption spectra (OA) by the formation of four peaks located at 263 nm, 257 nm, 252 nm, and 247 nm (wavelength difference of 5–7 nm between adjacent peaks over the absorption band) [3]. These peak-like formations resulted from the presence of quantum-confined dipeptide nanostructures [11,23]. In Figure 2a, the normalized OA spectrum of Boc-Phe-Phe-Bz in an ethanol solution with a concentration of 2.5 mg/mL shows an absorption band in the range 250 nm–350 nm, with three step-like peaks located at 290 nm, 299 nm, 304 nm, and 308 nm and a wavelength difference of 5–9 nm between adjacent peaks over the absorption band, indicating the formation of quantum-confined structures.

The absorption band of the new dipeptide ranged from 255 nm to 345 nm, while that of Boc-Phe-Phe was narrower (250 nm–300 nm). Figure 2b shows the normalized emission spectra for both dipeptides. For Boc-Phe-Phe, the maximum emission occurred at 283 nm for an excitation wavelength of 257 nm, while, for Boc-Phe-Phe-Bz, the maximum emission occurred at 369 nm for an excitation wavelength of 299 nm, with a red shift of 86 nm. Moreover, the emission spectra of Boc-Phe-Phe-Bz lay in the blue range of the optical spectra, with an intensity circa 30%, stronger than that measured for Boc-Phe-Phe.

As expected, the substitution in the Boc-Phe-Phe molecule of the OH group by a benzothiazole group greatly increased the blue fluorescence of the dipeptide. Interestingly, we observed a less intense emission band at 701 nm, already in the red part of the optical spectra. Table 1 summarizes these results.

The fluorescence characterization of Boc-Phe-Phe-Bz in an ethanol solution at a concentration of 1.6 × 10^−5^ M showed a broad emission spectrum in the visible light region between 320 and 500 nm when excited in the range between 280 and 370 nm (Figure 2c). Notably, a plot of the emission maximum as a function of the corresponding excitation wavelength indicated a consistent red shift in the fluorescence peak as the excitation wavelength increased. This phenomenon, known as the red-edge excitation shift (REES), suggested the presence of a dense, inhomogeneous energy landscape and has previously been observed in Dip-Dip nanocrystals, where Dip refers to dipeptide β,β-diphenyl-Ala-OH [17]. As depicted in Figure 2d, a strong linear correlation was observed between the emission maximum and the excitation wavelength, with a slope of approximately 0.63 nm/nm, slightly higher than the ~0.59 nm/nm reported for Dip-Dip crystals [17]. Furthermore, at longer excitation wavelengths, clear evidence of multiple peak emissions emerged, as previously reported [17]. This ability to tune the emission by simply adjusting the excitation wavelength presents promising opportunities for photonic and sensing applications. For example, REES has been demonstrated to be highly sensitive to tryptophan composition, making it a valuable tool for studying protein environments [24].

As shown in Figure 3, the absorption and emission studies of Boc-Phe-Phe-Bz in HFP/ultra-pure water (1:1) further reinforced the correlation between the optical properties and the nanostructure morphology. The absorption spectrum, Figure 3a, reveals four step-like peaks located at 236 nm, 268 nm, 293 nm, and 335 nm, characteristic of the formation of quantum-confined structures [25]. This observation mirrored the above findings for the dipeptide in an ethanol solution. Notably, in the HFP/ultra-pure water mixture, these peaks spanned a broader wavelength range and occurred at different positions compared to the ethanol solution. Additionally, a shoulder at 413 nm in the absorption band likely corresponded to larger nanostructures absorbing at higher wavelengths. Emission studies (Figure 3b,c), conducted over the 280–370 nm range demonstrated a consistent red shift of the fluorescence peak with increasing excitation wavelength. This phenomenon indicated an inhomogeneous energy landscape, similar to observations for the ethanol solution and those previously reported for Dip-Dip nanocrystals [17]. Once again, a strong linear correlation between the emission peak and the excitation wavelength was observed, with a slope of approximately 0.77 nm/nm. This value exceeded both the 0.63 nm/nm observed for the dipeptide in ethanol solution and the ~0.59 nm/nm reported for Dip-Dip nanocrystals [17].

The higher polarity of the HFP/ultra-pure water system promoted spectral shifts towards longer wavelengths, as would be expected for the self-assembly of larger nanoarchitectures, specifically nanobelts. These findings underscore the role of the solvent in modulating the optical properties of Boc-Phe-Phe-Bz. The observed differences between ethanol and HFP highlight the complex interplay between the solvent environment, nanostructure formation, and resulting optical characteristics, emphasizing the importance of considering solvent effects in the design and application of such dipeptide systems.

### 2.3. Scanning Electron Microscopy of Self-Assembled Nanostructures

SEM images of Boc-Phe-Phe-Bz crystallized in two different solvents were taken. It was found that there were different processes of self-assembly that were solvent-dependent. In particular, from an ultra-pure water/ethanol (1:1) solution, the dipeptide self-assembled into nanospheres with average diameter of around 579 nm, as shown in Figure 4a,b; when in a HFP/ultra-pure water (1:2) solution, it self-assembled into two different types of nanostructures: nanospheres with a smaller average diameter of 274 nm compared to those formed in the ultra-pure water/ethanol system (Figure 4c,d) and nanobelts with an average diameter of 36 nm, as shown in Figure 4e,f.

These findings highlight the remarkable ability of the new Boc-Phe-Phe-Bz system to modify the morphology of the formed nanostructures based on the solvents used.

Similar behavior has been reported for Boc-Phe-Phe, where supramolecular structures under the form of nanospheres and nanotubes have been reported to demonstrate self-assembling processes [26].

### 2.4. Dynamic Light Scattering Studies

Dynamic Light Scattering (DLS) is an analytical technique that makes use of the natural Brownian motion of particles in a fluid. This method allows for the evaluation of the hydrodynamic diameter of a sample by measuring the size of particles as they move through a liquid medium and the polydispersity in a non-destructive manner [27].

The hydrodynamic diameter of the particles was calculated using the Stokes–Einstein equation, assuming spherical particles:$$D=\frac{k_{B}T}{6\pi \eta d_{H}}$$
where $k_{B}$ is the Boltzman’s constant, T is the temperature, $d_{H}$ is the hydrodynamic diameter, *η* is the solvent viscosity, and D is the diffusion coefficient.

In a DLS experiment, a laser beam was directed at the sample and the scattered light was collected at a specific angle. The intensity fluctuations of the scattered light were analyzed to evaluate the diffusion behavior of the particles. These fluctuations were linked to particle size, with smaller particles diffusing more rapidly and generating faster intensity variations than larger particles. Polydispersity represents the width of the Gaussian distribution of particle sizes. It offers insight into the size variability of particles within a sample solution [28]. While SEM measures the geometric size of individual structures, DLS measures their hydrodynamic diameter, which includes the particle core and any associated hydration layer or adsorbed molecules [29]. However, there is an intensity bias in DLS, as larger particles or aggregates can dominate the signal, potentially masking smaller particles [30].

In this study, for Boc-Phe-Phe-Bz in an EtOH/water solution, the hydrodynamic radius measured by DLS (see Table 2) was of the same order of magnitude as the mean diameter obtained via SEM (see Figure 4a,b). Concordantly, as observed in Figure 5a, there appeared to be two distinct populations of particles with different diameters: a minor population with a diameter of approximately 66 nm and a predominant population of nanospheres with an average diameter of 541 nm.

For the second sample, Boc-Phe-Phe-Bz in the HFP/water solution, SEM analysis revealed the formation of two types of nanostructures: nanospheres and nanobelts (see Figure 4c,e). As shown in Figure 5b, DLS measurements indicated the presence of a larger population with hydrodynamic diameters peaked at 222 nm within the same order of magnitude as those obtained with SEM analysis (272 nm) for nanospheres and a minor population with diameters in the micrometer range, which corresponded to nanobelts formed from the agglomeration of smaller nanospheres.

The dipeptide nanostructures revealed a negative zeta potential of −30 mV, indicating a moderate electrostatic repulsion, which prevented rapid aggregation and enhanced the stability of the dispersion.

### 2.5. Nonlinear Optical Response

The second-order nonlinear optical response of Boc-Phe-Phe-Bz confirmed the presence of a noncentrosymmetric crystalline structure in the sample. Figure 6a shows a typical second harmonic spectrum. The inset displays the sample area illuminated during second harmonic generation (SHG) measurements. The bright, saturated spot on the right corresponds to the incident laser beam, while the spot on the left is a reflection from the second surface of the beam splitter used to capture the image with a CCD camera. Two additional, weaker reflections from the back surface of the microscope slide supporting the nanobelts were also visible above the primary spots. The dependence of the integrated SHG signal on the incident laser polarization is depicted in Figure 6b. For the data shown, the analyzer was adjusted to be aligned with the maximum signal.

Using a 1 mm thick BBO (β-Barium Borate, EKSMA) crystal cut for phase-matching at a fundamental wavelength of 800 nm, we calibrated the detection efficiency of our experimental setup. Under the strong focusing condition created by the ×10 microscope objective, the effective second harmonic generation (SHG) length within the BBO crystal was limited to approximately 35 µm due to spatial walk-off, which was 68 mrad for the SHG wavelength. The calibration procedure followed a previously reported method [31], and the relevant parameters and final results are summarized in Table 3.

To estimate the effective nonlinear susceptibility of the Boc-Phe-Phe-Bz nanobelts, it was essential to determine the effective thickness of the illuminated nanostructures. The CCD image inset in Figure 6a indicated that the laser spot covered an area comprising 3–4 overlapping nanobelts, with 2–3 aligned along a common diagonal orientation and one more vertically oriented. For an estimate of $d_{eff}$, we assumed a scenario of five overlapping nanobelts, each with a thickness of 40 nm, slightly above the mean value from the thickness distribution histogram of Figure 4f, resulting in a total thickness of ~200 nm. Given a nanobelt width of approximately 1 µm, which corresponded to the upper range of the hydrodynamic radius distribution, we obtained an estimated $d_{eff}$ of ~0.9 pm/V. There are very few studies of SHG measurements on dipeptide crystals; however, we could compare our result with that of 0.34 pm/V reported for triphenylalanine nanobelts, which are morphologically like Boc-Phe-Phe-Bz dipeptide. The response obtained for the new dipeptide was substantial for an organic crystal, further supporting that Boc-Phe-Phe-Bz crystallizes in a non-centrosymmetric structure [32].

## 3. Materials and Methods

### 3.1. Materials

2-Amino-6-ethoxybenzothiazole, L-Phenylalanine (Phe), 1-hydroxybenzotriazole (HOBt), *N*,*N*-dicyclohexylcarbodiimide (DCC), thionyl chloride, and di-tert-butylpyrocarbonate (Boc_2_O) were purchased from Merck/Sigma-Aldrich (Darmstadt, Germany), Alfa Aesar, and TCI (all purchased from Cymit Química, Barcelona, Spain, the distributor for both brands) and used as received. The solvents employed in this study, including dichloromethane (DCM), 1,1,1,3,3,3-hexafluoro-2-propanol (HFP), ethanol, methanol, and *N*,*N*-dimethylformamide (DMF), were purchased from Merck/Sigma-Aldrich. 1,4-dioxane was purchased from Fisher Chemicals (Zurich, Switzerland). All solvents were used as received, without further purification.

### 3.2. Procedure for the Synthesis of Tert-Butyl ((R)-1-(((R)-1-((6-ethoxybenzo[d]thiazol-2-yl)amino)-1-oxo-3-phenylpropan-2-yl)amino)-1-oxo-3-phenylpropan-2-yl)carbamate (Boc-Phe-Phe-Bz)

The Boc-protected diphenylalanine (Boc-Phe-Phe), synthesized according to the procedure described previously [3,33] (1.0 equiv), was dissolved in anhydrous dichloromethane (DCM), and the solution was cooled to 0 °C using an ice bath. *N*,*N’*-dicyclohexylcarbodiimide (DCC, 1.2 equiv) and 1-hydroxybenzotriazole hydrate (HOBt, 1.2 equiv) were added to the reaction mixture, which was stirred for 30 min at 0 °C. Subsequently, 2-Amino-6-ethoxybenzothiazole (Bz-NH_2_, 1.0 equiv) was introduced, and the mixture was stirred at room temperature for 48 h. Dichloromethane was then added, and the resulting solution was washed with water. The organic phase was dried over sodium sulfate (Na_2_SO_4_), filtered, and concentrated and the final residue was purified via column chromatography on silica gel, using mixtures of DCM/methanol with increasing polarity. The NMR spectra were performed on a Bucker spectrometer operating at 400 MHz and 100.6 MHz for ^1^H and ^13^C, respectively. The solvent used to prepare the samples was DMSO-*d*_6_ (dimethylsulfoxide >99.80 atom% D). Chemical shifts were reported in parts per million and tetramethylsilane (TMS) was used as an external reference. ^1^H NMR (400 MHz, DMSO-*d*_6_, ppm) δ 1.27 (3 × CH_3_-Boc, s, 9H), 1.32 (CH_3_-OEt, t, 3H), 2.48–2.49 (CH_2_-β, m, 1H), 2.67–2.72 (CH_2_-β, m, 2H), 3.10–3.14 (CH_2_-β, m, 1H), 4.02–4.05 (CH_2_-OEt, m, 2H), 4.19–4.20 (CH-α, m, 1H), 4.84–4.86 (CH-α, m, 1H), 6.81–6.84 (NH-Boc, d, 1H), 6.99–7.02 (CH-Bz, dd, *J_o_* = 8.8 Hz, and *J_m_* = 2.4 Hz, 1H), 7.13–7.32 (10 × CH-Phe, m, 10H), 7.52–7.53 (CH-Bz, d, *J_m_* = 2.8 Hz, 1H), 7.61–7.63 (CH-Bz, d, *J_o_* = 8.8 Hz, 1H), 8.32–8.34 (NH-amide, d, 1H), 12.44 (NH-Bz, s, 1H); ^13^C NMR (100.6 MHz, DMSO-*d*_6_, ppm): δ 14.73, 28.16, 54.26, 55.69, 63.68, 78.22, 105.43, 115.36, 121.28, 126.66, 128.02, 128.25, 129.24, 129.35, 132.89, 136.92, 137.95, 142.61, 155.17, 155.51, 155.58, 170.79, 171.94.

### 3.3. Self-Assembly of Dipeptide Nanostructures in HFP/Water and Ethanol/Water Solutions

Two solutions of Boc-Phe-Phe-Bz were prepared by dissolving the dipeptide in HFP/ultra-pure water (1:1) or ethanol/ultra-pure water (1:1) to concentrations of 5.7 mM and 4.3 mM, respectively. The solutions were left at room temperature for 24 h to allow self-assembly to occur. After that, a few drops of the Boc-Phe-Phe-Bz solutions were placed on silica slides, and the solvent was removed by slow evaporation at room temperature before the samples were subjected to SEM analysis.

These solutions were then further diluted in ultra-pure water to the desired final concentration of 5 µM according to the DLS study, with final solvent proportions of 0.1/99.9%.

### 3.4. Optical Absorption and Fluorescence

The 2.5 mg/mL solutions of Boc-Phe-Phe and Boc-Phe-Phe-Bz were subjected to optical absorption measurements using a Shimadzu UV-3101PC UV–vis–NIR spectrophotometer (Shimadzu, Kyoto, Japan). A Fluorolog 3 spectrofluorimeter (HORIBA Jobin Yvon IBH Ltd., Glasgow, UK) was used for fluorescence measurements. For optical absorption measurements, the dipeptide solutions were prepared in ethanol and HFP/ultra-pure water (both 1:1). These samples were introduced into a quartz cuvette with a 1 cm light path. For fluorescence (PL) measurements, spectra were recorded at a wavelength of 360–900 nm. Excitation wavelengths of 258 nm and 299 nm were used for Boc-Phe-Phe and Boc-Phe-Phe-Bz, respectively, with fixed entrance and exit slits to obtain a spectral resolution of 2 nm.

### 3.5. Dynamic Light Scattering (DLS)

To determine the hydrodynamic size and polydispersity index (PDI) of the Boc-Phe-Phe-Bz nanostructures, dynamic light scattering (DLS) analysis was performed using a Litesizer 500 from Anton Paar. This equipment features three detection angles (15°, 90°, and 175°) and utilizes a semiconductor laser diode with a wavelength of 658 nm and a power output of 40 mW. Measurements were conducted in a quartz cell at room temperature with a backscatter detection angle, with each sample analyzed three times. The experimental data were processed using Kalliope software (version 1.8) to obtain the hydrodynamic size and PDI values.

For DLS, the sample was diluted in ultra-pure water to a final concentration of 5 µM, treated in an ultrasonic bath, and filtered to 0.45 µm before analysis.

### 3.6. Scanning Electronic Microscopy (SEM)

The morphology, diameter distribution, and thickness of the dipeptide nanostructures were obtained using a Nova NanoSEM scanning electron microscope at an accelerating voltage of 10 kV. Dipeptide single crystals were deposited on a silica surface and previously covered with a thin film (10 nm thickness) of Au-Pd (80–20 weight %) using a high-resolution sputter cover, 208HR Cressington Company, coupled with an MTM-20 Cressigton high-resolution thickness controller. The diameter range of the produced nanofibers was measured from SEM images using ImageJ 1.51n image processing software (NIH, https://imagej.nih.gov/ij/, 6 November 2024). The average diameter and diameter distribution were determined by measuring a certain number of random nanofibers from the SEM images. Statistical analysis was performed using OriginPro 2017 SR2 software (OriginLab Corporation, Northampton, MA, USA) and fiber diameter distributions were fitted to the log-normal function.

### 3.7. Differential Scanning Calorimetry (DSC) and Thermogravimetric Analysis (TGA)

Differential scanning calorimetry (DSC) analysis was performed in a Netzsch 200 Maya (Netzsch, Selb, Germany) under a nitrogen flow (50 mL/min). The sample was placed in an aluminum pan. A first heating ramp was performed from 20 °C to 120 °C, cooled down to 20 °C, and heated again up to 200 °C (second heating) at 2 °K/min. Thermogravimetric analysis (TGA) was performed using a TA Q500 thermogravimetric analyzer (TA Instruments, New Castle, DE, USA). The sample was placed in a platinum crucible and heated from 30 °C to 700 °C at a heating rate of 10 °C/min under a nitrogen flow (60 mL/min).

### 3.8. Second Harmonic Generation

The second harmonic signal was generated using a mode-locked Ti/sapphire oscillator (Coherent Mira, Coherent, Inc., Saxonburg, PA, USA) with a nominal pulse duration of 100 fs and a repetition rate of 76 MHz. The experimental setup was essentially the same as that previously used [31]. Briefly, incident power on the sample was controlled using a combination of a half-wave plate and polarizer. The laser beam was focused onto the sample using a ×10 microscope objective (Plan Fluorite, Nikon, Osaka, Japan) with a numerical aperture of 0.30 and an effective focal length of 20 mm, producing a focal spot with a diameter of approximately 5 µm at the $1/e^{2}$ intensity level. A set of chirped mirrors from Edmund Optics precompensated the pulse so that the nominal pulse duration was 110 fs (FWHM). The transmitted second harmonic signal was collected using a second ×10 objective (Plan Achromat, Olympus, Tokyo, Japan) and passed through a dichroic mirror, followed by a zero-order half-wave plate and a fixed calcite polarizer. The signal was then filtered with a low-pass cut-off filter (transition wavelength of 650 nm) and focused into a multimode fiber bundle. At the fiber output, a 0.3 m imaging spectrometer (Shamrock 303i, Andor, Belfast, UK) was used to isolate the second harmonic signal near 400 nm, which was subsequently captured by a cooled CCD camera (Newton 920, Andor, Belfast, UK) and integrated over the second harmonic spectrum. To control both the incident and transmitted linear polarization states, zero-order half-wave plates were positioned before the focusing objective (800 nm) and after the collimating objective (400 nm), respectively. The polarization curve in Figure 6b corresponds to the transmitted polarization state that maximized the detected second harmonic signal. By rotating the half-wave plate before the focusing objective, the polarization of the incident fundamental light was varied over two full revolutions to measure the dependence of the second harmonic signal on the incident polarization angle.

## 4. Conclusions

In conclusion, we designed and synthesized a new chiral and aromatic-rich dipeptide, *N-tert*-butoxycarbonyl-protected diphenylalanine benzothiazole, which self-assembles in different supramolecular structures, specifically as nanospheres and nanobelts, in different solvents.

These self-assembled architectures display increased blue fluorescence due to the presence of the bicyclic ring benzothiazole group. Furthermore, a consistent red shift in the fluorescence peak as the excitation wavelength increased was observed, accompanied by a strong linear correlation between the emission maximum and the excitation wavelength.

Finally, the new dipeptide nanobelts, which crystallize in a non-centrosymmetric structure, show a good nonlinear optical response, with an estimated $d_{eff}$ of ~0.9 pm/V, higher than that of 0.34 pm/V reported for triphenylalanine nanobelts.

## Figures and Tables

**Figure 1 molecules-30-00942-f001:**
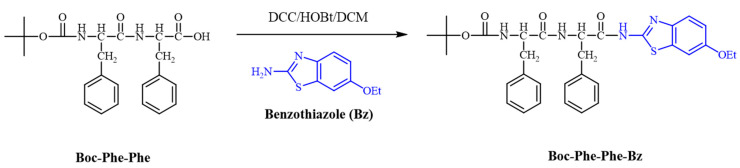
Schematic description of the synthesis of Boc-Phe-Phe-Bz dipeptide.

**Figure 2 molecules-30-00942-f002:**
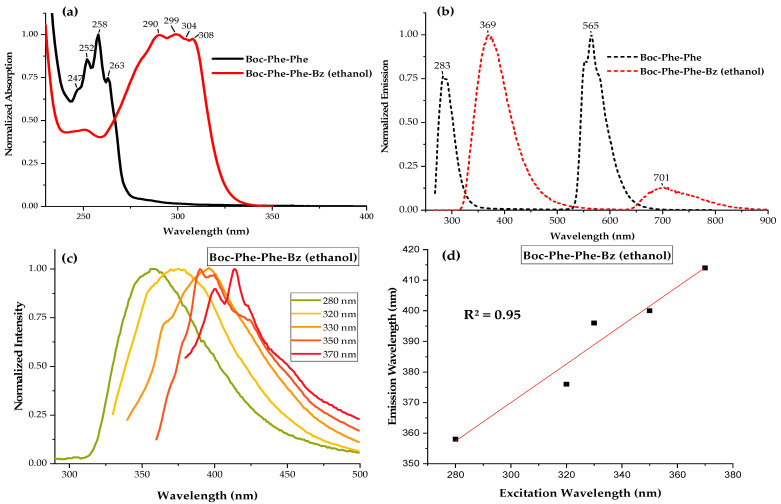
(**a**) Normalized optical absorption spectra (solid line) in ethanol at 4.25 × 10^−3^ M and (**b**) fluorescence spectrum at 2.5 × 10^−5^ M (dashed line) of Boc-Phe-Phe and Boc-Phe-Phe-Bz, measured at room temperature. (**c**) Normalized fluorescence spectra of Boc-Phe-Phe-Bz in ethanol with a concentration of 1.6 × 10^−5^ M at different excitations ranging between 280 and 370 nm. (**d**) Emission versus excitation wavelengths, with a linear fit. The slope of the best-fit straight line was 0.63 nm/nm.

**Figure 3 molecules-30-00942-f003:**
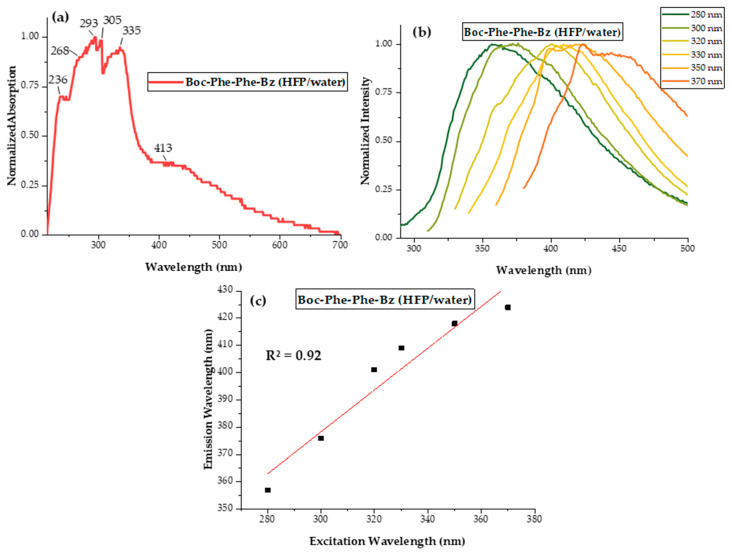
(**a**) Normalized optical absorption spectrum of Boc-Phe-Phe-Bz in HFP/ultra-pure water (1:1) at a concentration of 3.2 × 10^−3^ M. (**b**) Normalized fluorescence spectra of Boc-Phe-Phe-Bz in a mixture of HFP/ultra-pure water (1:2) with a concentration of 8.5 × 10^−5^ M at different excitations ranging between 280 and 400 nm. (**c**) Emission versus excitation wavelengths with a linear fit. The slope of the best-fit straight line was 0.77 nm/nm.

**Figure 4 molecules-30-00942-f004:**
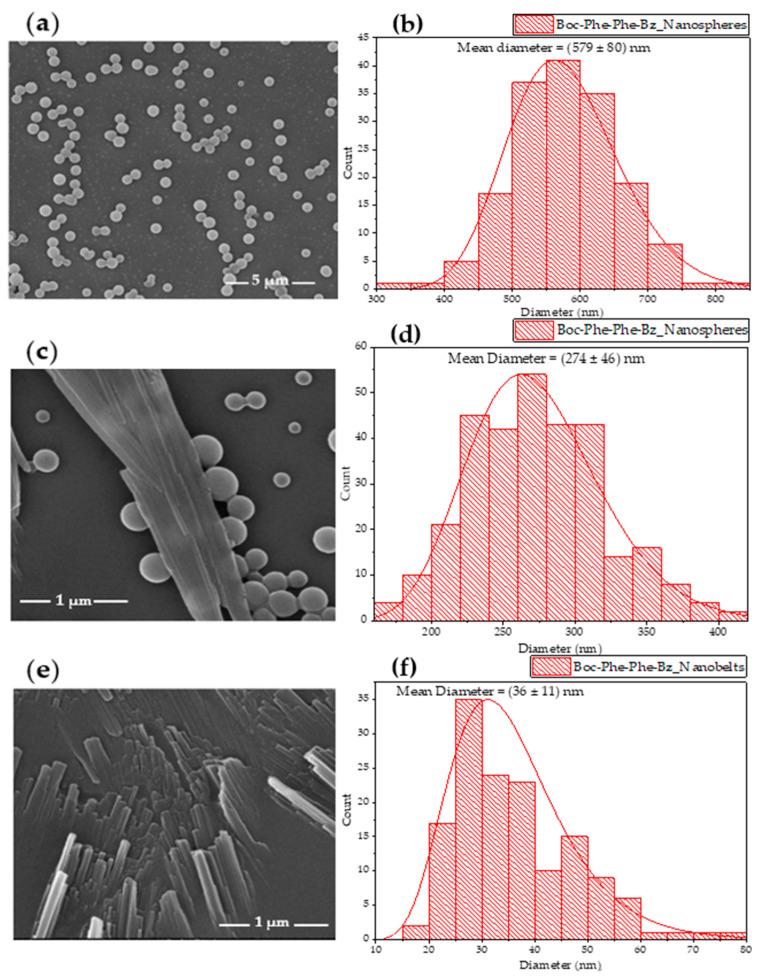
(**a**,**b**) SEM images of Boc-Phe-Phe-Bz dipeptide nanospheres at 15,000× magnification and their respective size distribution histograms in ethanol/ultra-pure water (1:1) system. (**c**–**f**) SEM images of Boc-Phe-Phe-Bz dipeptide nanospheres and nanobelts at 100,000× magnification and their respective size distribution histograms in HFP/ultra-pure water (1:2) system. The red curves indicate lognormal distributions.

**Figure 5 molecules-30-00942-f005:**
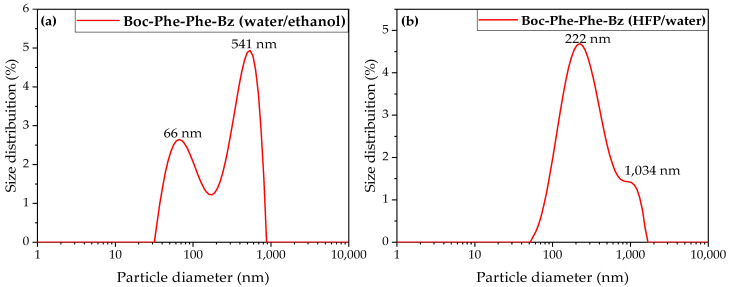
Intensity-weighted particle size distributions of Boc-Phe-Phe-Bz with a concentration of $5\times 10^{-6}$ M in (**a**) ultra-pure water/ethanol and (**b**) HFP/ultra-pure water.

**Figure 6 molecules-30-00942-f006:**
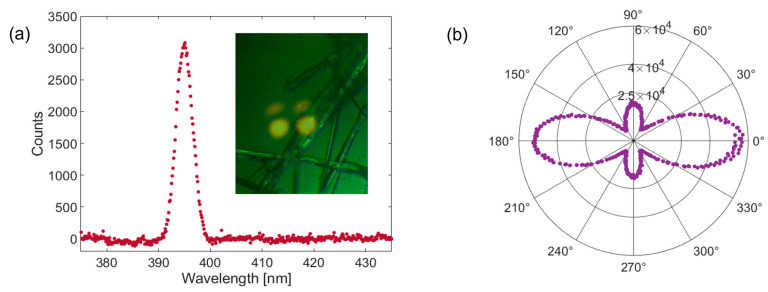
(**a**) A representative second harmonic spectrum. The inset is an image of the illuminated region responsible for generating the second harmonic signal. The bright spot on the right corresponds to the incident laser beam. (**b**) Polarization dependence of the second harmonic signal, with the analyzer adjusted to maximize the detected intensity.

**Table 1 molecules-30-00942-t001:** Maximum absorption and emission wavelengths of Boc-Phe-Phe and Boc-Phe-Phe-Bz in an ethanol solution.

Dipeptide	λ_abs_ [nm]	λ_emi_ [nm]
Boc-Phe-Phe	247; 252; 258; 263	283; 565 [3]
Boc-Phe-Phe-Bz	290; 299; 304; 308	369; 701

**Table 2 molecules-30-00942-t002:** Hydrodynamic mean diameter, transmittance, polydispersity index (PDI), and diffusion coefficient of the Boc-Phe-Phe-Bz nanospheres diluted in ultra-pure water/ethanol (0.1/99.9%) and nanobelts in HFP/ultra-pure water (0.1/99.9%) obtained by DLS.

Boc-Phe-Phe-Bz	Hydrodynamic Diameter [nm]	Transmittance [%]	Polydispersity Index [%]	Diffusion Coefficient
Nanospheres	482	99	30	1.02
Nanobelts/nanospheres	380	99	25	1.28

**Table 3 molecules-30-00942-t003:** Second harmonic generation parameters for Boc-Phe-Phe-Bz.

Sample	Fundamental Wave Average Power (mW)	Signal Integration Time (ms)	Effective Thickness (µm)	Second Harmonic Signal (Counts)	$d_{eff}$ (pm/V)
BBO	0.62	0.1	35	6.5 × 10^6^	2.0
Boc-Phe-Phe-Bz	62	20	0.20	5.7 × 10^4^	0.9

## Data Availability

The original contributions presented in this study are included in the article/Appendix A. Further inquiries can be directed to the corresponding authors.

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
