# Peer review of "Novel Benzothiazole Boc-Phe-Phe-Bz Derivative Dipeptide Forming Fluorescent and Nonlinear Optical Self-Assembled Structures"

_molecules, 2025, doi:10.3390/molecules30040942_

Round 1

Reviewer 1 Report

Comments and Suggestions for Authors

I have read with interest the paper of R.M.F. Baptista et al. on a new dipeptide derivative and its optical and other properties. The paper is non-standard as it describes the properties not often discussed in chemical papers (e.g., DLS analysis). I think the paper deserves acceptance as may cause the interest of the readers and prompt some of them to pay attention to studying similar or related properties of their own objects.

I have no comments to the reviewed paper that would require any serious revision, except for one small remark for the SI:

Fig. SI1: α-CH (not α-CH2) in captions to signals. 

Author Response

Comments:

I have read with interest the paper of R.M.F. Baptista et al. on a new dipeptide derivative and its optical and other properties. The paper is non-standard as it describes the properties not often discussed in chemical papers (e.g., DLS analysis). I think the paper deserves acceptance as may cause the interest of the readers and prompt some of them to pay attention to studying similar or related properties of their own objects.

I have no comments to the reviewed paper that would require any serious revision, except for one small remark for the SI: Fig. SI1: α-CH (not α-CH2) in captions to signals.

Response: Thank you, we agree and have corrected it.

Reviewer 2 Report

Comments and Suggestions for Authors

The manuscript presents an innovative and highly relevant study on the self-assembly and optical properties of a novel chiral Boc-Phe-Phe dipeptide derivative. The integration of the benzothiazole bicyclic ring into the dipeptide structure is a notable advancement, significantly enhancing fluorescence and enabling tunable supramolecular morphologies. The authors provide a comprehensive analysis, from the molecular design to the nonlinear optical properties, which underscores the potential of these structures in advanced optical applications. The work is well-structured, with clear experimental methodologies and impactful results that contribute meaningfully to the field of supramolecular chemistry and materials science.

However, before it can be accepted for publication, some points must be clarified, and authors need to address the following main issues:

1.       It is recommended to add references that support and deepen the influence of chirality on the formation process of self-assembling aggregates, such as: 10.1016/j.bioorg.2021.105047 , 10.3389/fbioe.2021.703004 (lines 27-29).

2.       Consider expanding on the role of Phe-Phe in self-assembly processes to highlight their versatility. In addition to forming well-ordered structures such as nanotubes, nanowires, and nanorods (lines 36-39), these dipeptides also play a significant role in the formation of nanoparticles under specific conditions as supported by recent literature (doi: 10.1038/s41598-024-60145-z.)

3.       The manuscript mentions that SEM, DLS and optical studies are also performed using HFIP/water as solvents, whereas fluorescence is only observed in ethanol. It would be helpful to explain why fluorescence is not also performed in HFP/water. therefore, could the authors provide more information on why fluorescence is exclusive to ethanol? Discussing the potential impact of solvent polarity or other properties on fluorescence behaviour would improve the understanding of these observations.

4.       Please amplify the resolution of all the figures inserted in the supplementary information.

5.       Regarding SEM images, in my opinion it would be better to insert the scale bar with a contrasting color on the black background (maybe white) rather than leave it in red.

6.       Do the authors have any idea about the charge of the nanospheres (Zeta-potential)? Furthermore, the authors make no reference to the population of 66 nm particle sizes observed in the case of Figure 5 panel A.

7.       Are these nanospheres stable over time or/and in different fluids? Furthermore, what can be the possible applications?

8.       Please, check the typo “s” in line 153 in the sentence: “…. ability of the new s Boc-Phe-Phe-Bz ”

Author Response

Comments: The manuscript presents an innovative and highly relevant study on the self-assembly and optical properties of a novel chiral Boc-Phe-Phe dipeptide derivative. The integration of the benzothiazole bicyclic ring into the dipeptide structure is a notable advancement, significantly enhancing fluorescence and enabling tunable supramolecular morphologies. The authors provide a comprehensive analysis, from the molecular design to the nonlinear optical properties, which underscores the potential of these structures in advanced optical applications. The work is well-structured, with clear experimental methodologies and impactful results that contribute meaningfully to the field of supramolecular chemistry and materials science.

However, before it can be accepted for publication, some points must be clarified, and authors need to address the following main issues:

Dear Reviewer: Thank you very much for your comments and suggestions. We are very grateful to you for taking your time reading our manuscript.

Comment 1. It is recommended to add references that support and deepen the influence of chirality on the formation process of self-assembling aggregates, such as: 10.1016/j.bioorg.2021.105047 , 10.3389/fbioe.2021.703004 (lines 27-29).

Response 1: Thank you, we have included them in line 28.

Comment 2. Consider expanding on the role of Phe-Phe in self-assembly processes to highlight their versatility. In addition to forming well-ordered structures such as nanotubes, nanowires, and nanorods (lines 36-39), these dipeptides also play a significant role in the formation of nanoparticles under specific conditions as supported by recent literature (doi: 10.1038/s41598-024-60145-z.)

Response 2: Thank you, we have included the sentence in lines 41- 43.

Comment 3. The manuscript mentions that SEM, DLS and optical studies are also performed using HFIP/water as solvents, whereas fluorescence is only observed in ethanol. It would be helpful to explain why fluorescence is not also performed in HFP/water. therefore, could the authors provide more information on why fluorescence is exclusive to ethanol? Discussing the potential impact of solvent polarity or other properties on fluorescence behaviour would improve the understanding of these observations.

Response 3: Thank you, we have conducted fluorescence studies in HFP/water (1:1) and integrated the corresponding results into the revised manuscript. New figures have been included to illustrate these findings, along with a detailed discussion on the influence of solvent polarity and other relevant properties on the fluorescence behaviour. We thank the reviewer for this valuable suggestion, which has helped to improve the completeness of our study.

Comment 4. Please amplify the resolution of all the figures inserted in the supplementary information.

Response 4: Thank you, we have increased the resolution of all figures in supplementary information.

Comment 5. Regarding SEM images, in my opinion it would be better to insert the scale bar with a contrasting color on the black background (maybe white) rather than leave it in red.

Response 5: Thank you, we have changed the scale bar to the suggested colour.

Comment 6. Do the authors have any idea about the charge of the nanospheres (Zeta-potential)? Furthermore, the authors make no reference to the population of 66 nm particle sizes observed in the case of Figure 5 panel A.

Response 6: Thank you, from SEM measurements, we conclude that the population of particles with 66 nm size are also nanospheres. As written in the manuscript, there is a minor population with a diameter of approximately 66 nm and a predominant population of nanospheres with an average diameter of 541 nm. The dipeptide nanostructures revealed a negative zeta potential, -30 mV, indicating a moderate electrostatic repulsion, which prevents rapid aggregation and enhances the stability of the dispersion. We have included this information in the manuscript document, line 233.

Comment 7. Are these nanospheres stable over time or/and in different fluids? Furthermore, what can be the possible applications?

Response 7: Thank you, we have not tried to put these nanospheres in different fluids. However, it might be a good suggestion for future research featuring its applications in biotechnology.

Comment 8. Please, check the typo “s” in line 153 in the sentence: “…. ability of the new s Boc-Phe-Phe-Bz ”

Response 8: Thank you, that mistake has been corrected.

Reviewer 3 Report

Comments and Suggestions for Authors

Dear Authors,

The manuscript topic is interesting and it has been well written. I have only some minor reviews that I listed in the file attached. I suggested writing in the introduction the difference between this manuscript and the previously reported and cited as 1. In this publication, you prepared the same compound. I also reported something about the carbon NMR, figure SI2 of Supporting Informations.

Author Response

Comment 1. In the introduction authors must write that they previously synthesized the compound that has been studied in this paper.1 They have also write why they repeated the synthesis and the different results that they obtained and are described in this manuscript.

Response: Thank you, the compound presented in this manuscript is a new dipeptide. The article you are referring to is a preprint of this submission as explained below.

Comment 2. I found the paper with the same title and same authors by google search. I don’t understand why: DOI:10.20944/preprints202501.1206.v1

Response: Thank you, the paper you found is a preprint version, which was uploaded to a preprint server, at the time we submitted to MOLECULES the present manuscript. This is not a duplicate publication but a common practice to share research before formal peer review. 

Comment 3. Probably it should be useful to report the optical properties of benzotriazole, authors can measure using the starting material of the synthesis (1-hydroxybenzotriazole hydrate) or find in the literature if reported.

Response: Thank you for your comment. HOBt (1-hydroxybenzotriazole) and DCC (dicyclohexylcarbodiimide) are commonly used as coupling reagents in peptide synthesis in solution. HOBt acts as an additive to prevent racemization and improve the efficiency of amide bond formation, while DCC activates the carboxyl group by forming an O-acylisourea intermediate. Since HOBt is only used as a coupling agent/intermediate in our synthesis, its optical properties are not relevant to this study. Regarding the benzothiazole moiety coupled to the dipeptide, its optical properties, particularly fluorescence, are already discussed in the introduction, including how they contribute to the fluorescence of the dipeptide after HOBt/DCC coupling.

Comment 4. Row 35: [1-6] after “interest.”

Response: Thank you for your comment. We have corrected this in the manuscript.

Comment 5. Row 34: “Their rich hydrogen bonding network together with directional intermolecular π–π interactions originates their self-assembling with quantum nanocrystalline regions under the form of quantum dots with strong quantum confinement” in which solvent?

Response: The solvents are ethanol and HFP. Reference [10.1002/psc.96] was added in the manuscript, lines 45, 47 and 98.

Comment 6. Row 74: authors must specify who is DCC (I think dicyclohexylcarbodiimide), HOBt (I think hydroxybenzothiazole hydrate)

Response: Thank you, we have correctly specified DCC and HOBt in the text.  

Comment 7. Row 80-84: it should be good to provide experimental mass obtained by mass spectrometry for full characterization

Response: Thank you. Unfortunately, we do not have mass spectrometry data, but the proton and carbon NMR spectra confirm the success of the synthesis, showing all expected peaks.  

Comment 8: Figure SI2 13CNMR: in the spectrum it is indicated only one quaternary carbon of benzothiazole at 172 ppm, but this molecular group contains 4 quaternary carbons that I cannot see in the spectrum. It is also strange that the quaternary carbon of tert-butyl ester is at 105 ppm, it should be at around 80 ppm.

Response: Thank you. It is not always possible to assign all peaks in the carbon spectrum, and since quaternary carbon signals are of lower intensity, they may overlap with other peaks. The signal at 105 ppm is unlikely to correspond to a quaternary carbon, as its intensity is higher than expected. We will remove this assignment from the spectrum.

Round 2

Reviewer 2 Report

Comments and Suggestions for Authors

The authors fulfilled all the requirements, so I suggest publication of this manuscript.